# Discovery of Drugs Targeting Mutant p53 and Progress in Nano-Enabled Therapeutic Strategy for p53-Mutated Cancers

**DOI:** 10.3390/biom15060763

**Published:** 2025-05-26

**Authors:** Na Zhang, Zhiyuan Jing, Jie Song, Qiyue Liang, Yuxue Xu, Zhaowei Xu, Longping Wen, Pengfei Wei

**Affiliations:** 1Shandong Technology Innovation Center of Molecular Targeting and Intelligent Diagnosis and Treatment, School of Pharmacy, Binzhou Medical University, Yantai 264003, China; zhangna20@bzmc.edu.cn (N.Z.); 2022082600@stu.bzmc.edu.cn (Z.J.); 2023081004@stu.bzmc.edu.cn (J.S.); 2023081039@stu.bzmc.edu.cn (Q.L.); xuyuxue@bzmc.edu.cn (Y.X.); zhaoweixv@bzmc.edu.cn (Z.X.); 2Guangdong Provincial People’s Hospital (Guangdong Academy of Medical Sciences), Southern Medical University, Guangzhou 510080, China

**Keywords:** mutant p53, mutant p53 reactivation, protein stability, nanodelivery

## Abstract

Mutations in the p53 gene are frequently observed in various cancers, prompting the initiation of efforts to restore p53 function as a therapeutic approach several decades ago. Nevertheless, only a limited number of drug development initiatives have progressed to late-stage clinical trials, and to date, no p53-targeted therapies have received approval in the USA or Europe. This situation can be attributed primarily to the characteristics of p53 as a nuclear transcription factor, which lacks the conventional features associated with drug targets and has historically been considered “undruggable”. In recent years, however, several promising strategies have emerged, including the enhanced iterations of previous approaches and novel techniques aimed at targeting proteins that have traditionally been considered undruggable. There is a growing interest in small molecules that can restore the tumor-suppressive functions of mutant p53 proteins, and the development of drugs specifically designed for particular p53 mutation types is currently underway. Other approaches aim to deplete mutant p53 or exploit vulnerabilities associated with its expression. Additionally, genetic therapy strategy and approaches have rekindled interest. Advances in mutant p53 biology, compound mechanisms, treatment modalities, and nanotechnology have opened up new avenues for p53-based therapies. However, significant challenges remain in clinical development. This review reassesses the progress in targeting p53-mutant cancers, discusses the obstacles in translating these approaches into effective therapies, and highlights p53-based therapies via nanotechnology.

## 1. Introduction

The p53 gene, identified in 1979 [1], is a crucial tumor suppressor with diverse functions in cancer biology. Initially mischaracterized as interacting with viral proteins, it was later recognized for its role in inhibiting tumor growth. Early research on p53 involved synthesizing cDNA from tumor cell mRNA, initially suggesting its role as an oncogene that enhances tumorigenesis [2]. However, later studies revealed that the p53 protein in normal cells functions as a tumor suppressor, effectively inhibiting the proliferation of tumor cells [3]. The gene encodes a 43.7 kDa protein, but its migration anomaly in electrophoresis led to its naming as p53 [4]. Over time, p53 has been recognized as a critical transcription factor regulating cell cycle arrest, apoptosis, and DNA repair, while also exerting antiproliferative effects through non-transcriptional mechanisms [5]. It influences nearly all cellular compartments, encompassing lysosomes, mitochondria, and the endoplasmic reticulum [6]. Beyond tumor suppression, p53 plays roles in metabolic homeostasis, immune regulation, and tumor cell stemness [7]. With over 120,000 publications, p53 has emerged as the most extensively researched gene in recent decades, playing a pivotal role in the fields of molecular biology and oncology.

In cells containing wild-type p53 (wtp53), excessively high levels of p53 under non-stressed conditions may negatively impact the growth and development of normal cells. Upon activation by various stimuli, p53 transcribes the MDM2 gene, leading to increased levels of the MDM2 mRNA and protein. The MDM2 protein directly binds to p53 through its N-terminal domain and suppresses p53 levels and function via three primary mechanisms: (1) MDM2 ubiquitinates p53 through its E3 ligase activity, promoting its degradation via the proteasome pathway. Interestingly, MDM2 mediates p53 monoubiquitination, while p300 facilitates subsequent polyubiquitination [8]; (2) the interaction between MDM2 and p53 obstructs p53’s binding to its target DNA, thereby preventing p53 from functioning as a transcription factor; (3) MDM2 facilitates the nuclear export of p53 [9,10], preventing p53 from accessing its target DNA and further diminishing its transcriptional capacity (Figure 1A–C).

Mutations in the tumor suppressor gene p53 (TP53) represent the most prevalent genetic alterations observed in human cancers, occurring in over 50% of tumors and up to 80% in aggressive cancers like high-grade serous ovarian cancer and small-cell lung cancer [11,12]. Most p53 mutations are classified as missense mutations, which can be categorized into two distinct types: DNA-contact mutations (e.g., R273H and R248Q), which prevent DNA binding, and conformational mutations (e.g., R175H, Y220C, R249S), which disrupt the protein structure [13]. Over 80% of these mutations are localized within the DNA-binding domain (DBD), leading to a loss of tumor suppressor function and a gain of oncogenic properties [14]. Mutant p53 (mutp53) often exhibits oncogenic gain-of-function (GOF), linked to increased tumor aggressiveness, reduced patient survival, and resistance to standard therapies [13]. While most mutp53 proteins cannot bind DNA or activate wtp53 tumor suppressor pathways, they can disrupt wtp53 function by forming tetramers and exerting a dominant-negative (DN) effect [15]. The high mutation frequency of p53 in cancers, coupled with its inherent role in suppressing wtp53, makes it a promising therapeutic target [11]. When p53 undergoes mutations, it affects MDM2’s recognition of it and its interaction with DNA. First, when the hydrophobic core of p53 is disrupted (for example, by the R175H mutation) [16], the mutation at this site leads to the unwinding of the α-helix structure, causing a change in the spatial orientation of the W23 side chain, which weakens the binding to MDM2; or its phosphorylation sites may be interfered with (for example, by the G245D mutation) [17], preventing MDM2 from recognizing it. Secondly, mutations may lead to a loss of functionality at the direct contact sites of p53 (for example, the R248Q mutation) [18] or a loss of conformational stability (for example, the R175H mutation) [19], making it unable to form a stable DNA-binding surface, thereby reducing its binding ability to DNA (Figure 1A–D).

For decades, drug development targeting p53 has seen limited progress, leading to the characterization of p53 as an ‘undruggable’ target [20]. Challenges arise from the smooth surface of mutp53 proteins and the lack of a suitable drug-binding pocket, complicating efforts to restore its tumor-suppressive function. However, with technological advancements, many previously undruggable targets, such as KRAS, are now becoming druggable [21,22], offering hope for the development of p53-targeted drugs. Recent innovations have opened new avenues for p53-based therapies. Strategies include screening or designing molecules to reactivate mutp53 to restore its tumor-suppressive function [23], utilizing protein degradation technologies such as proteolytic-targeting chimeras (PROTAC) to degrade mutp53 [24], leveraging nanotechnology for targeted drug delivery [25], etc. Additionally, recent in-depth investigations into the mechanisms that maintain the stability of mutp53 proteins [26,27], including our findings [28], have also provided critical breakthroughs for therapies targeting mutp53. These advancements underscore the growing viability of p53 as a target for therapeutic intervention.

To advance research on mutp53-targeted cancer therapies, it is crucial to systematically compile the latest anticancer studies, analyze the role of mutp53 in tumor signaling pathways, and elucidate recent progress in targeting mutp53 for treatment. This review reassesses the progress in targeting p53-mutant cancers, discusses the obstacles in translating these approaches into effective therapies, and highlights mutp53-based cancer therapies via nanotechnology.

## 2. Advances in mutp53-Targeting Drug Discovery

### 2.1. Compounds Reactivating mutp53

Reactivating mutp53 is challenging, but studies show that mutp53 can refold into its wild-type conformation under certain conditions, such as low temperatures or through peptides derived from its C-terminal domain [29]. The p53 protein exhibits sensitivity to heat and is capable of undergoing conformational changes between its folded and unfolded states as a result of mutations. This raises the possibility of using small molecules to reverse mutp53’s conformation and restore its wild-type function. High-throughput screening has identified a few compounds capable of reactivating mutp53, enabling it to induce apoptosis, promote cell cycle arrest, and inhibit cancer cell proliferation. This “mutant reactivation” phenomenon offers promising therapeutic potential for targeting mutp53 in cancer. As reviewed previously [6,30], some small-molecule drugs that have been identified as capable of restoring the wild-type conformation and/or mutate the DNA-binding capacity of p53 include CP31398, PRIMA-1, APR-246 (also named PRIMA-1MET or eprenetapopt), PC14586, arsenic trioxide (ATO), NSC319726 (also known as ZMC1), COTI-2, MIRA-1, STIMA-1, RETRA, PK1107, NSC59984, ReACp53, HO-3867, ADH-6, PhiKan083, PK7088, and others (Figure 2A,B). Notably, CP31398 is recognized as the first small-molecule compound capable of activating the conversion of mutp53 to the wtp53 conformation [6], thereby stabilizing the activity of wtp53 and facilitating tumor suppression. However, due to issues related to solubility and toxicity to normal cells, only a limited number of drugs targeting mutp53 have progressed to clinical trials. The compounds presently under clinical investigation include APR-246, ATO, PC14586, and COTI-2.

APR-246: Pharmacologically, PRIMA-1 acts as a prodrug, with its degradation product, MQ, covalently reacting with the thiol groups of cysteine residues in the core domain of mutp53, thereby restoring the wtp53 conformation [31]. Additionally, MQ can alter cellular redox balance by directly binding to cysteine in glutathione (GSH) and inhibiting thioredoxin reductase, disrupting key homeostatic mechanisms essential for cell survival and promoting cancer cell death [32]. Over time, several other drugs aimed at reactivating mutp53 have been discovered. For instance, both MIRA-1 and STIMA-1 exhibit Michael acceptor activity similar to PRIMA-1 [33,34]. These compounds can modify cysteine residues in the p53 protein, thereby stabilizing its wild-type conformation and preventing the aggregation of mutp53. Despite demonstrating p53-dependent effects in vitro and in vivo, neither MIRA-1 nor STIMA-1 have advanced to clinical trials due to issues such as poor solubility (STIMA-1) and toxicity to normal cells (MIRA-1) [6]. PRIMA-1 MET, a methylated derivative of PRIMA-1, also referred to as APR-246, has shown superior activity compared to PRIMA-1. Interestingly, APR-246 does not induce apoptosis in cells harboring the p53 (Y220C) hotspot mutation [35], indicating that its efficacy is mutation-specific and not universal across all p53 mutants. Beyond its effectiveness as a monotherapy, APR-246 has been shown to augment cancer cell death when used in conjunction with chemotherapy. Two pivotal clinical trials (NCT03072043 and NCT03588078) demonstrated promising efficacy when APR-246 was combined with azacitidine (the first FDA-approved drug for myelodysplastic syndrome (MDS) in patients with TP53-mutated MDS or acute myeloid leukemia (AML) [36,37]. The combination regimen showed favorable tolerability with a safety profile comparable to either agent used as a monotherapy. Further supporting evidence comes from a completed phase III trial (NCT03745716) evaluating APR-246 plus azacitidine in TP53-mutant MDS [38]. While the combination improved the complete remission rates (34.6% vs. 22.4% with azacitidine alone), this benefit was possibly offset by increased all-cause mortality (58.97% vs. 47.37%). Notably, a phase II study (NCT03931291) investigating this combination as post-transplant maintenance therapy in TP53-mutated AML/MDS patients demonstrated encouraging relapse-free survival (RFS) and overall survival (OS) outcomes in this high-risk population, with good tolerability [39]. The phase I/II trial (NCT04214860) evaluating frontline APR-246 combined with AZA and VEN in TP53-mutated AML met its primary efficacy endpoint, demonstrating a CR/CRi rate of 53% (CR 37%) in preliminary results [40]. This triple therapy shows promising potential for TP53-mutated AML. The therapeutic potential of APR-246 extends beyond hematologic malignancies, as demonstrated in a phase Ib/II trial (NCT02098343) involving patients with platinum-sensitive, TP53-mutated high-grade serous ovarian cancer. The study revealed that the addition of APR-246 to carboplatin and pegylated liposomal doxorubicin (PLD) significantly improved the complete response (CR) rate (9.5% vs. 2.8% with carboplatin–PLD alone). However, this enhancement in deep responses was accompanied by a slightly lower overall disease control rate. While the combination therapy showed promise in driving deeper remissions, the higher progressive disease (PD) rate suggests the need for biomarker-driven patient selection to optimize clinical benefit. Notably, this improved efficacy came at the cost of increased toxicity, with the incidence of adverse events nearly doubling (31.31% vs. 16.83%). A phase II clinical trial (NCT03268382) evaluated the efficacy of APR-246 combined with PLD in patients with platinum-resistant recurrent high-grade serous ovarian cancer (HGSOC) harboring TP53 mutations. The combination therapy demonstrated a disease control rate (DCR) of 69.6%, while the incidence of grade ≥3 adverse events was 39.29%. These clinical advancements have not only validated the clinical druggability of the previously “undruggable” target TP53 but also revealed three promising directions: the need to develop predictive biomarkers for patient stratification, the optimization of combination strategies, and balancing efficacy–toxicity profiles through novel drug delivery technologies, collectively paving a new paradigm for p53-associated malignancy therapeutics.

ATO: ATO, a traditional Chinese medicine, has demonstrated clinical efficacy in the treatment of acute promyelocytic leukemia, with response rates exceeding 80% when administered as a monotherapy and between 70% and 80% when used in conjunction with chemotherapy all-trans retinoic acid, as evidenced by a phase IV study (NCT00504764). The therapeutic efficacy of ATO is primarily attributed to its ability to facilitate the degradation of the PML-RARα fusion protein [41,42]. Additionally, ATO has been shown to induce apoptosis in tumor cells carrying TP53 mutations, as reviewed previously [30]. Recently, Chen et al. discovered through computational analysis that ATO can rescue the function of structural p53 mutants and stabilize their conformation but has minimal effects on DNA-contact mutants [43]. ATO demonstrates the different degrees of functional recovery capabilities for 390 common p53 structural mutants, particularly temperature-sensitive ones [44]. The researchers further administered standard-dose (0.16 mg/kg/day) ATO monotherapy to a non-APL AML patient carrying an ATO-sensitive p53 mutation, observing the effective restoration of p53 transcriptional regulatory function accompanied by the significant clearance of minimal residual disease (MRD) [44]. Although previous clinical observations have suggested ATO’s potential therapeutic effects in various solid tumors, and this study represents the first in vivo demonstration of ATO’s functional restoration of p53 mutants, these findings still require validation through prospective, randomized controlled clinical trials to confirm whether ATO exerts its antitumor effects via p53 functional recovery. Notably, ATO’s clinical application faces several key challenges: (1) The drug’s inherent toxicity limits its therapeutic window. (2) The prevalence of p53 mutations fully restorable by ATO is merely 0.05–0.1% in AML and only 0.48% across pan-cancer types. (3) As an approved drug, ATO-related research is mostly limited to investigator-initiated trial models [44]. These factors collectively pose significant challenges to patient recruitment [44]. Currently, several clinical trials investigating ATO for p53-mutated malignancies are underway, though most have not disclosed their progress (Table 1). Of particular interest is an ongoing phase II clinical trial (NCT06088030) evaluating the safety and efficacy of ATO in treating pediatric malignancies with p53 mutations, which has commenced patient recruitment.

PC14586: During the screening process for compounds aimed at reactivating the p53 (Y220C) mutant, the developed PC14586 has entered the clinical trial phase [6]. PC14586 is an orally bioavailable small molecule specifically designed to target the Y220C mutation. PC14586 demonstrated a manageable safety profile and preliminary antitumor activity in a phase I/II trial (NCT04585750) involving 29 patients with TP53 Y220C-mutant solid tumors. Treatment-related adverse events (79% incidence, mostly grade 1–2) included nausea (34%), vomiting (24%), fatigue (21%), and aspartate aminotransferase level elevation (17%), with no dose-limiting toxicities. Among twenty-one evaluable patients, five achieved partial responses [45]. The results support the further development of PC14586 in this molecularly defined population at doses of up to 3000 mg/day. A phase I b clinical trial (NCT06616636) evaluating PC14586 in combination with azacitidine for TP53 Y220C-mutant myeloid M = malignancies (acute myeloid leukemia or myelodysplastic syndromes) is currently enrolling participants, with clinical outcomes yet to be reported. While the initial success of Y220C-targeted therapies such as PC14586 is encouraging, it is important to note that the Y220C mutation is relatively rare, and its distinct structural features differ from other p53 mutations. Therefore, employing analogous structure-based strategies for other p53 mutants may pose greater challenges, highlighting the need for innovative approaches to target the broad spectrum of p53 mutations.

Thiosemicarbazones: Zinc ions (Zn^2+^) play a critical role in stabilizing the DBD of the p53 proteins, promoting their proper folding and transcriptional function [46]. However, mutp53 exhibits reduced affinity for Zn^2+^, leading to protein instability, misfolding, and aggregation, which in turn contribute to tumorigenesis. The structural mutant p53 (R175) serves as a classic example of this phenomenon. Studies have shown that increasing the Zn^2+^ concentration could stabilize wtp53 and activate its signaling pathway [25]. Through the National Cancer Institute’s 60 human tumor cell line anticancer drug screening, the zinc metallochaperone-1 (ZMC1) was identified, which selectively targets p53(R175H), inducing apoptosis and tumor regression in p53(R175H)-expressing cells while sparing wtp53 cells [47]. The activity of ZMC1 depends on its Zn^2+^-chelating properties and redox changes. However, ZMC1 has limited Zn^2+^-binding capacity and undergoes rapid in vivo metabolism, potentially causing off-target toxicity (e.g., neurotoxicity due to zinc overload). To enhance zinc affinity for a more precise Zn^2+^ release in the tumor microenvironment (TME) and improve p53 reactivation capabilities, optimized chemical structures yielded related compounds, ZMC2 and ZMC3, two other thiosemicarbazones, which deliver Zn^2+^ more efficiently and also restore p53(R175H)’s wild-type conformation [48]. Nevertheless, ZMCs only apply to Zn^2+^-sensitive p53 mutations, and cancer cells may evade treatment by reducing Zn^2+^ uptake. To develop drugs that bind more stably to p53, reduce resistance, and cover a broader spectrum of p53 mutations, COTI-2 was identified as a third-generation thiosemicarbazone through a comprehensive computational platform. It shows activity in both p53-mutant and wild-type cells, acting through p53-dependent and p53-independent pathways, which include p63-mediated DNA damage [49]. Among these thiosemicarbazone compounds, only COTI-2 has entered phase I clinical trials (NCT02433626) [50], while the other R175H-targeted compounds remain in preclinical stages. These findings highlight the potential of reactivating conformational p53 mutants, though further optimization is needed for clinical use.
biomolecules-15-00763-t001_Table 1Table 1Summary of molecules reactivating mutp53 in clinical trials. The clinical trial data and identification numbers were sourced from the Clinical Trials.gov website of the U.S. National Library of Medicine, which can be searched and retrieved via the link: https://clinicaltrials.gov/ (accessed on 10 May 2025).CompoundsInterventionCancer TypeExperimental or Clinical ResultsPhase StatusClinical Trial Identifier or CitationsAPR-246APR-246 + azacitidineP53-mutated AML or MDS following allogeneic stem cell transplantOne-year non-recurrence survival rate of 59.9% and a median overall survival of 20.6 months; well tolerated.Phase II, completedNCT03931291; [39]APR-246 + azacitidineP53-mutant myeloid neoplasmsThe overall remission rate for 55 patients was 71%, with 44% achieving complete remission. The frequency of TP53 mutation alleles and p53 expression were significantly reduced.Phase I b/II, completedNCT03072043 [36]APR-246 + azacitidineP53-mutant myeloid neoplasmsCompared to azacitidine monotherapy, it demonstrated a favorable safety profile and showed potentially superior ORR, complete response (CR) rate, and overall survival (OS).Phase II, completedNCT03588078 [37]APR-246 + azacitidineP53-mutant MDSCompared to azacitidine, the complete remission rate increased from 22.4% to 34.6%.Phase III, completedNCT03745716 [6]APR-246 + venetoclax + azacitidineP53-mutant myeloid malignanciesCompared to the combination of APR-246 and venetoclax, an 8.91% reduction in serious adverse events and a 39.53% reduction in all-cause mortality.Phase I, completedNCT04214860 [40]APR-246 + carboplatin and pegylated liposomal doxorubicin hydrochloridePlatinum-sensitive recurrent high-grade serous ovarian cancer (HGSOC) with mutated p53CR: 9.5%.Partial responses (PRs): 40.0%.SD: 25.7%.Phase I b/II, completedNCT02098343APR-246 + pegylated liposomal doxorubicin hydrochloridePlatinum-resistant recurrent high-grade serous ovarian cancer (HGSOC) with mutated TP53The ORR is 69.6%, and the incidence of severe adverse reactions is 39.29%.Phase II, completedNCT03268382APR-246 + venetoclaxRelapsed refractory mantle cell lymphoma patients (with/without p53 aberrations)Not disclosed.Phase II, withdrawnNCT04990778APR-246, 5-FU, and cisplatinPlatinum-resistant advanced and metastatic oesophageal cancers/Phase I b/II, terminatedNCT02999893ATOATOAcute myeloid leukemia with p53 mutationsNot disclosed.Phase II, unknown statusNCT03381781ATOP53-mutated pediatric cancerNot disclosed.Phase II, recruitingNCT06088030ATOPreviously untreated or relapsed/refractory p53-mutated myeloid malignanciesNot disclosed.Phase II, not yet recruitingNCT06778187ATO + decitabine, intravenouslyAML/MDS-expressing mutant p53Not disclosed.Phase I, unknown statusNCT03855371PC14586PC14586 + pembrolizumabLocally advanced or metastatic solid tumors harboring a p53 Y220C mutation21 efficacy-evaluable patients (out of 29 total); PRs were observed in 5 cases; treatment-related adverse events occurred in 79% of patients.Phase I/II, recruitingNCT04585750[45]PC14586 + azacitidineTP53 Y220C-mutant AML/MDSNot disclosed.Phase I b, recruitingNCT06616636; [51]COTI-2COTI-2 monotherapy and COTI-2/cisplatin combination therapyAdvanced or recurrent malignanciesNot disclosed.Phase I, unknown statusNCT02433626


### 2.2. Targeting mutp53 Protein Stability

The strategy of degrading mutp53 utilizes the high-level accumulation and survival-dependent characteristics of cancer cells toward mutp53. Under normal circumstances, wtp53 exhibits low expression levels in cells as a result of degradation mediated by the ubiquitin–proteasome system (UPS). The regulation of the wtp53 protein half-life is critical: in normal tissues, wtp53 is rapidly degraded primarily through MDM2-mediated ubiquitination, but it remains stable under stress conditions. In contrast, cancer cells harboring TP53 are characterized by the accumulation of high levels of mutp53 proteins, which is a central feature of its GOF properties. Although the mutp53 protein typically accumulates at high levels in tumor cells [52], its expression in normal tissues can be maintained at low levels via MDM2 [53,54]. This observation suggests that key determinants within the TME or tumor cells themselves are instrumental in facilitating the stability and accumulation of mutp53. Additionally, different mutp53 alleles may possess unique attributes and engage in interactions with various proteins [55]. To date, the mechanisms underlying mutp53 protein accumulation may involve multiple pathways (Figure 3), which provide alternative avenues to combating p53-mutated cancers via interfering with mutp53 stability.

One viewpoint suggests that tumor-associated stress may trigger the binding of mutp53 to molecular chaperones, leading to the stability of mutp53 (Figure 2B, Table 2). For example, the use of HSP90 inhibitors (e.g., Ganetespib) or HDAC inhibitors (HDACis) can cause cytotoxicity in tumor cells and enhance their sensitivity to chemotherapy drugs [14,56]. The mechanism behind it is that HSPs can maintain the stability of the mutp53 protein by preventing the interaction with E3 ubiquitin ligases, including MDM2 or CHIP (Figure 3). The HDAC inhibitor vorinostat, also referred to as suberoylanilide hydroxamic acid (SAHA), promotes the degradation of mutp53 through the inhibition of HDAC6, which acts as a positive regulator of HSP90 (Figure 3). This inhibition facilitates the degradation of mutp53 by MDM2 and CHIP [56]. Statins or mevalonate kinase knockdown can reduce mevalonate-5-phosphate, disrupting the interaction between structural mutp53 and DNAJA1, which is a member of the Hsp40 family. This disruption makes mutp53 more susceptible to recognition by E3 ubiquitin ligases like MDM2 or CHIP, leading to its ubiquitination and proteasomal degradation [57,58]. These findings highlight DNAJA1 as a key regulator of misfolded mutp53 stability, suggesting the mevalonate pathway–DNAJA1 axis as a promising therapeutic target (Figure 3). Ingallina et al. [59] discovered that the accumulation of HDAC6/HSP90-dependent mutp53 is maintained by RhoA geranylgeranylation downstream of the mevalonate pathway, as well as by the RhoA- and actin-dependent transduction of mechanical inputs. The inhibition of the mevalonate pathway (such as using statins, e.g., cerivastatin), abrogating RhoA geranylgeranylation (e.g., with GGTI-298), or RhoA activity (e.g., with C3 toxin) can destabilize mutp53, reducing its oncogenic effects and suppressing cancer cell proliferation and metastasis [59]. Statins, with their established safety and pharmacology, are particularly suitable for repurposing in p53-mutated cancer treatment. A phase II clinical study (NCT02012192) evaluated the efficacy and safety of Ganetespib combined with paclitaxel in patients with metastatic p53-mutant platinum-resistant ovarian cancer. The results showed that, compared to paclitaxel alone, the combination therapy did not demonstrate superiority in all-cause mortality (with a slight increase observed) and was associated with a higher incidence of severe adverse events, including anemia, sepsis, small-bowel obstruction, and gastrointestinal reactions, such as diarrhea. Preliminary data from a phase I trial (NCT01339871) demonstrated that the pazopanib (an anti-angiogenic drug pazopanib)–SAHA combination showed promising activity in TP53-mutant solid tumors, with hotspot-mutant patients (*n* = 11) achieving a 45% disease control rate, with significantly longer progression-free survival (PFS) and mOS compared to patients with undetected hotspot TP53 mutations (*n* = 25). Adverse reactions included hematologic toxicities (such as thrombocytopenia and neutropenia), as well as fatigue, hypertension, etc. However, through optimized dose adjustment, this treatment regimen still warrants further investigation into TP53-mutant metastatic sarcomas and colorectal cancers [60]. However, further investigation is required to ascertain whether these clinical benefits are present and if they are attributed to the degradation of mutp53.

The further elucidation of the key mechanisms underlying the accumulation of mutp53 in cancer provides new insights for the targeted degradation of mutp53. Jiang et al. [61] demonstrate that malic enzyme 2 (ME2) stabilizes mutp53 by regulating 2-hydroxyglutarate (2-HG) synthesis, which binds mutp53 and reduces MDM2-mediated degradation, a mechanism specific to mutp53 but not wtp53 (Figure 3). Additionally, TRIM21, an E3 ubiquitin ligase, down-regulated expression in colorectal cancer and breast cancer, and related to prognosis, selectively ubiquitinates and degrades mutp53, suppressing its GOF effects [26]. However, further research is needed to explore how to modulate TRIM21 activity or expression for therapeutic benefits. Zoltsman et al. [27] underscores the dual role of Class A JDPs in maintaining protein homeostasis and promoting cancer progression through the stabilization of mutp53. By targeting the unique β-hairpin region of Class A JDPs, it may be possible to disrupt the stabilization of mutp53, offering a novel therapeutic avenue for cancers driven by p53 mutations. Recently, we also highlight the role of the reducing status surrounding the mutp53 proteins in stabilizing mutp53 by counteracting S-glutathionylation and ubiquitination (Figure 3). Inhibiting GSH reductase (e.g., by 2-AAPA) or depleting GSH promotes mutp53 but not wtp53 degradation via the proteasome [28]. In addition, disulfiram non-selectively could promote the degradation of both wtp53 and mutp53 proteins by inducing protein glutathionylation modifications [62]. In contrast, our developed 2-AAPA selectively degrades mutp53 proteins through GSH reductase inhibition and demonstrates superior growth inhibitory effects on mutp53 tumors. This discovery provides a novel research direction for developing mutp53-targeted cancer therapies. These insights provide promising avenues for mutp53-targeted cancer therapies.

Autophagy, a lysosome-dependent degradation process, maintains protein and organelle stability in eukaryotic cells [63]. Mutp53 could disrupt autophagy, leading to its accumulation and oncogenic effects, as reviewed previously [64]. Vice versa, inhibiting autophagy could stabilize mutp53, suggesting autophagy induction as a potential therapeutic strategy for mutp53 tumors. Mutp53 often aggregates in hypoxic tumor cores. Strategies like glucose restriction [65], CHIP overexpression [66], curcumin-based zinc compounds (the structure of the compound was deduced and is presented in Figure 2A) [67,68], gambogic acid [69], spautin-1 [70], and an FDA-approved histone deacetylase inhibitor (SAHA) [71] enhance mutp53 clearance (Figure 2A,B, Table 2). While research is in early stages and focused on specific p53 mutants, autophagy regulation remains a promising approach for targeting mutp53 in cancer therapy.
biomolecules-15-00763-t002_Table 2Table 2Summary of molecules targeting mutp53 protein stability. The clinical trial data and identification numbers were sourced from the Clinical Trials.gov website of the U.S. National Library of Medicine, which can be searched and retrieved via the link: https://clinicaltrials.gov/ (accessed on 10 May 2025).CompoundsMechanism of ActionStage of Development (Up to Now)Clinical DataCitationsHsp90 inhibitors (e.g., Ganetespib)Promote the degradation of mutp53 via inhibiting HSP90Phase I/II clinical trial/NCT02012192/terminatedp53-mutant platinum-resistant ovarian cancer: Ganetespib plus paclitaxel shows inferior safety profile compared to paclitaxel monotherapy.[14]HDACi (e.g., SAHA)Promote the degradation of mutp53 through the inhibition of HDAC6, which acts as a positive regulator of HSP90Phase I clinical trial/NCT01339871/terminatedInitial data warrant further investigation of this regimen in TP53-mutant cancers, especially metastatic sarcoma/colorectal cancer.[56,60,71]DisulfiramInduces glutathionylation of p53 and degradation of both wild-type and mutant p53sPhase I or II clinical trials for various cancers but not designed for p53-mutated tumors/[62]GSH reductase inhibitors (e.g., 2-AAPA)Selectively induce the proteasomal degradation of mutp53 via promoting glutathionylation and subsequent K48-linkage polyubiquitination of mutp53 proteinsPreclinical/[28]Gambogic acidPromotes the degradation of mutp53 (R280K and S241F) via inducing autophagyPreclinical/[69]Spautin-1Promotes the degradation of mutp53 via inducing chaperone-mediated autophagyPreclinical/[70]Curcumin-based zincPromotes the degradation of mutp53 (R175H) via inducing autophagyPreclinical/[67]


### 2.3. Synthetic Lethality

Mutations in the p53 gene are prevalent in a majority of malignancies, leading to deficiencies in DNA repair mechanisms at the G1/S checkpoint. TP53-mutated cells with replication errors may bypass the G1 checkpoint and advance to the G2 phase [13]. Consequently, disruptions in G2 and M checkpoints, critical for DNA replication and genome integrity, can severely impact cell viability [72,73]. Recent investigations have identified mutp53 as a primary focus in synthetic lethality research, demonstrating a “synthetic lethality” interaction with proteins such as WEE1, PKC, PLK1, PARP, ATM, ATR, and CHK1, which play vital roles in the regulation of the cell cycle and the repair of DNA damage [73,74,75,76,77,78]. Notably, WEE1 is an important kinase in the DNA damage repair pathway, and plays a critical role in regulating the G2 checkpoint, which is essential for the viability of cells with p53 mutations [13]. As a result, the WEE1 inhibitor adavosertib exhibits pronounced synthetic lethality in p53-altered contexts, effectively inhibiting tumor progression associated with p53 deletion or mutation [77]. Compared to drugs directly targeting mutp53, synthetic lethal approaches are less dependent on p53’s structure, making them applicable to a broad range of p53 mutations. This highlights the potential of synthetic lethality as a versatile strategy for targeting p53-mutant cancers (Figure 2B).

### 2.4. p53-Based Genetic Therapies

#### 2.4.1. Wtp53-Based Gene Therapies

Gendicine, recognized as the first gene therapy authorized for clinical application, is a p53-targeted treatment. In 2003, the China Food and Drug Administration (CFDA) granted approval for a recombinant human p53 adenovirus product intended for the treatment of head and neck squamous cell carcinoma (HNSCC). Gendicine, which has been utilized in clinical applications for over two decades, signifies a noteworthy advancement in the field of gene therapy. Qi et al. [78] conducted a comprehensive review of Gendicine’s biological mechanisms of action, clinical responses, administration protocols for various cancer types, and associated adverse reaction rates. Notably, clinical studies (ChiCTR-TRC-09000392, ChiCTR-TRC-08000094) have demonstrated that the combination of Gendicine with conventional radiotherapy and chemotherapy regimens may yield more substantial therapeutic outcomes in tumor growth suppression and patient prognosis improvement compared to radiotherapy and chemotherapy alone [79,80,81]. Other adenovirus-based p53 gene therapies have shown encouraging outcomes in clinical trials (NCT00002960); however, in contrast to gendicine, they have not yet received approval for clinical application [6]. It is important to note that significant concerns exist regarding the efficacy of this approach, particularly if the reconstruction of p53 fails to interact with additional therapeutic targets [6]. The advent of novel and more sophisticated viral vectors may enhance the effectiveness and accessibility of p53 gene therapy, potentially allowing it to be integrated into a combination therapy regimen.

#### 2.4.2. RNA Therapeutics

In comparison to traditional small molecule- and antibody-based therapies, RNA-based drugs possess a direct regulatory capability, providing unique advantages in the targeting of pathogenic genes and their associated proteins, which are challenging to address with conventional pharmacological approaches. RNA therapeutics can be classified according to their mechanisms of action into several categories, including antisense oligonucleotides (ASOs), RNA interference agents (such as siRNA and miRNA), mRNA therapeutics, and nucleic acid aptamers. Among these, ASO drugs are the most widely utilized form of RNA therapeutics [82]. The application of RNA therapy offers several significant advantages: (1) high specificity for targets, (2) modularity with easily modifiable sequences, (3) predictable pharmacokinetic and pharmacodynamic profiles, (4) cost-effectiveness compared to antibody- or protein-based therapies due to their synthesis from readily accessible oligonucleotides using automated synthesizers, and (5) a relatively favorable safety profile, as the majority of RNA drugs do not alter the genome [83]. However, significant challenges persist in the medicinal use of RNA drugs. Firstly, RNA molecules demonstrate poor stability, and unlike small molecules, RNA must penetrate cells to exert therapeutic effects. RNA is prone to degradation by nucleases found in biological environments, and unprotected RNA is prone to rapid degradation, complicating effective cellular uptake [84]. Secondly, the intrinsic properties of RNA drugs, such as their large molecular weight, size, and negative charge, hinder their ability to penetrate cell membranes and reach their designated sites of action [85]. Additionally, there exists a risk of off-target effects during the administration of RNA drugs, which may cause accumulation in non-target tissues, thereby diminishing therapeutic efficacy and potentially causing significant toxic side effects [86]. To mitigate these challenges, nanodelivery systems are utilized to encapsulate RNA drugs, thereby enhancing their stability and promoting efficient targeted delivery (Figure 4).

Recent advancements in mRNA-based therapies have addressed several challenges associated with traditional gene therapy and protein correction/restoration approaches. mRNA therapy primarily involves delivering mRNA into the cytoplasm of target cells. This method circumvents the need for mRNA to enter the nucleus, thereby reducing the risk of genomic integration, as mRNA can be directly translated into proteins within the cytoplasm. While mRNA can effectively express therapeutic proteins similar to therapeutic DNA, it exhibits a higher vulnerability to degradation by nucleases and is generally regarded as more immunogenic than DNA. However, by designing and customizing mRNA sequences in vitro, these sensitivities can be minimized. Despite these advantages, mRNA faces challenges in traversing negatively charged cell membranes due to electrostatic repulsion. Furthermore, once inside the cell, unprotected mRNA is easily degraded by nucleases and may trigger immunogenic responses. Consequently, the development of delivery vectors for mRNA is essential for enhancing its therapeutic efficacy.

In the realm of p53-targeted genetic therapy, the utilization of synthetic small interfering RNA (siRNA) oligonucleotides presents a promising avenue to counteract the GOF effects of mutp53 by specifically targeting mutations within p53 mRNA. The precision of siRNA in distinguishing between wtp53 and mutp53 through a single base difference was demonstrated, leading to reduced cancer cell survival and enhanced apoptosis upon targeting the p53(R248W) mutants [87]. Advancing this approach, Ubby et al. [88] developed siRNAs tailored to four distinct p53 hotspot mutants, showcasing their efficacy in diminishing the viability of patient-derived xenografts in a manner that is specific to the mutations, while also exhibiting no organ toxicity and not affecting the mRNA levels of wtp53. Despite the nascent stage of siRNA-based therapies, these discoveries, coupled with significant progress in RNA delivery technologies, underscore the potential for further investigation into mutant-specific p53 siRNA as a viable strategy for cancer treatment.

#### 2.4.3. CRISPR-Cas9

The advent of CRISPR-Cas9 technology has sparked significant enthusiasm, prompting exploration into its application for p53-targeted cancer therapy (Figure 4). In a study targeting HCC1954 breast cancer cells, a base editor was employed to revert a TP53 missense mutation to its wild-type sequence, achieving a 7.6% correction rate [89]. While the biological impact of this correction was not examined, the potential for the clinical application of TP53 mutation correction via base editing appears promising. However, CRISPR-Cas9 induces DNA damage that activates the p53 pathway in cells possessing wtp53, leading to cell cycle arrest or apoptosis and potentially favoring the survival of TP53-mutant cells [90,91,92]. Paradoxically, TP53-mutant cancer cells successfully converted to wtp53 might be more susceptible to elimination due to concurrent DNA damage signaling. A critical challenge lies in achieving successful conversion in the majority of cancer cells to ensure clinical efficacy. The potential of CRISPR-Cas9 TP53 base editing as a viable therapeutic option in humans remains to be seen, as its feasibility and effectiveness await further validation.

### 2.5. PROTAC

Recent advancements in targeted PROTAC have facilitated the degradation of such pathogenic proteins. PROTACs are bifunctional molecules that degrade target proteins by linking them to an E3 ubiquitin ligase, marking the proteins for proteasomal destruction. This approach targets ‘undruggable’ proteins, offers sustained degradation, and addresses drug resistance, showing promise for treating cancer and neurodegenerative diseases [24,93,94]. The PROTAC methodology has been employed to maintain wtp53 stability, mainly via targeting MDM2 for degradation, leading to wtp53 carrying tumor regression [95,96,97,98]. In fact, as long as a suitable ligand for the target protein is found, any specific protein, including mutp53, may be selectively cleared. Kong et al. [99] developed an RNA aptamer-based PROTAC (dp53m-RA) that selectively degrades the oncogenic mutant protein p53-R175H without affecting wtp53 or other p53 mutants. dp53m-RA degrades p53-R175H via the UPS and significantly inhibits the proliferation and migration of cancer cells expressing this mutant protein. They [100] also developed dp53m, a DNA aptamer-based PROTAC, to selectively degrade the p53-R175H mutation. Dp53m showed high specificity, stability, and no systemic toxicity, effectively inhibiting tumor growth and enhancing cisplatin sensitivity in p53-R175H-driven cancers. These discoveries offer a promising therapeutic approach for cancers harboring the p53-R175H mutation, demonstrating the potential of PROTACs as a targeted treatment for cancers with mutp53.

## 3. Nanostrategies for Targeting mutp53 for Cancer Therapy

Nanotechnology demonstrates significant advantages in cancer treatment, including high targeting efficiency (achieving tumor-specific enrichment through the enhanced permeability and retention effect and the surface modification of targeting molecules) [101,102], controlled drug release (enabling selective killing by responding to the TME) [103], multifunctional integration (combining combination therapy and theranostics) [104,105,106], reduced toxicity (minimizing damage to normal tissues) [107,108], overcoming drug resistance (reversing the oncogenic function of mutp53 and enabling multi-mechanism synergy) [109,110,111], and personalized treatment (precision medicine and real-time monitoring). These features make nanotechnology a powerful tool for treating mutp53-related tumors, offering new pathways for safer and more effective precision therapy.

### 3.1. Delivering mutp53-Reactivating Agents

Nanoparticles offer a promising strategy to deliver mutp53 inhibitors, reversing their oncogenic functions and enhancing cancer cell sensitivity to treatment (Table 3). For instance, C16-ceramide, which binds to the DBD of p53 and disrupts its interaction with the E3 ligase MDM2, elevates p53 levels and promotes tumor therapy [112]. Delivering C16-ceramide via ceramide–rubusoside (Cer-RUB) nanomicelles enhances its uptake and bioavailability, restoring wtp53 expression and increasing p21 levels in mutp53-expressing models, thereby improving therapeutic outcomes [113]. However, ceramide metabolism by glucosylceramide synthetase can limit its efficacy, suggesting that inhibiting this enzyme may further restore wtp53 function and induce apoptosis in mutp53-driven cancers [114,115,116]. Additionally, recombinant peptide nanoparticles (MtrapNPs) mimic ARF to sequester MDM2, stabilizing p53 and enhancing its tumor-suppressive effects while also delivering ATO as a salvage therapy for p53-mutant tumors [117]. In diffuse large B-cell lymphoma (DLBCL), combining p53 reactivation (via stapled peptide ATSP-7041) with BCL-2 inhibition (ABT-263) using CD19-targeted polymersomes has shown reduced toxicity and improved efficacy in preclinical models [118]. The albumin nanovectors (ANVs) have been successfully applied in the delivery of clinical drugs such as paclitaxel and have achieved clinical translation. Compared to traditional solvent-based formulations, they significantly enhance tumor targeting and improve drug safety. ANVs are also proposed for the targeted delivery of ZMC-1 to reactivate p53, offering a potential therapeutic approach for BRCA1-related breast cancers with p53 mutations and highlighting the versatility of nanoparticle-based strategies in addressing mutp53-driven cancers [119]. A thermosensitive gel–nano system is capable of selectively delivering siRNA targeting BACH1 (siBACH1) alongside the p53 activator PRIMA-1. This approach amplifies the inhibitory effect of siBACH1 on BACH1 expression, subsequently reinstating p53 activity and augmenting T cell immune responses [120].

### 3.2. Triggering mutp53 Clearance

#### 3.2.1. Nanodelivery of Zinc Ions

Studies have shown that increasing Zn^2+^ concentration also could induce the conformational transition of mutp53 to a wild-type-like state [46], promoting its degradation via the UPS or autophagy pathway [46,121,122]. However, Zn^2+^ itself is cytotoxic and difficult to internalize into cells, prompting the development of various zinc-loaded nanoparticles that are utilized to enhance the intracellular delivery and targeting of Zn^2+^ (Table 4). For example, ZnFe-4 nanoparticles continuously release Zn^2+^ in acidic endosomes, promoting reactive oxygen species (ROS) generation and mutp53 degradation [123]; the zeolitic imidazolate framework-8 (also denoted as ZIF-8, a kind of Zn^2+^-based metal–organic framework) induces mutp53 degradation by releasing Zn^2+^ and depleting GSH, with RGD peptide modification improving cancer cell targeting [124]; Mn-ZnO_2_ NPs release Zn^2+^ and Mn^2+^ in the acidic TME, promoting mutp53 degradation and wtp53 expression, respectively, while enhancing antitumor effects through the Fenton reaction-generated hydroxyl radicals [125]. Additionally, zinc-based dihydroxide nanosheets restore wtp53 function and induce mutp53 degradation, respectively, by releasing Zn^2+^ and glucose oxidase [126], while zinc-doped Prussian blue nanoparticles utilize photothermal effects to accelerate zinc-induced mutp53 degradation and tumor cell apoptosis [127]. ZIF-8@MnO_2_ nanoparticles release elevated concentrations of Zn^2+^ and Mn^2+^, which promote the degradation of the mutp53 protein through the UPS. This alleviates the inhibitory effect of mutp53 on the cGAS-STING pathway, subsequently enhancing immune responses. Additionally, they synergize with PD-L1 inhibitors to improve immunotherapy efficacy (Figure 5) [128]. These zinc-loaded nanoparticles promote mutp53 degradation and restore wtp53 function through multiple mechanisms, offering novel strategies for treating p53-mutant tumors.

#### 3.2.2. Delivering Compounds Capable of Clearing mutp53

Targeting mutp53 through advanced nanocarrier systems has emerged as a promising strategy to overcome drug resistance and enhance cancer therapy (Table 4). PEITC-modified black phosphorus nanosheets (BPNs) integrate phenylethyl isothiocyanate (PEITC) to degrade mutp53 and reverse multi-drug resistance, while their near-infrared light-responsive properties enable photothermal therapy, inhibiting tumor growth and facilitating controlled drug release with reduced toxicity to healthy tissues [129]. Currently, there are no FDA-approved pharmacological agents specifically targeting mutp53-driven cancers, but repurposing existing clinical drugs with established safety profiles offers a promising avenue for research. For example, crizotinib, an FDA-approved drug, has been encapsulated in PEG-PLGA nanocapsules to enhance tumor-specific delivery and minimize systemic toxicity, selectively degrading mutp53 via the UPS without affecting wtp53 (Figure 6) [130]. Cisplatin, a primary therapeutic agent for non-small-cell lung cancer (NSCLC), often leads to TP53 mutations, contributing to cisplatin resistance [131]. To address this, fluoroplatin-PE nanoparticles (FP NPs) were developed by combining cisplatin with statins, such as rosuvastatin, and modifying the surface with poly(ethylene glycol)-phosphoethanolamine to enhance cellular uptake and promote preferential accumulation in the endoplasmic reticulum and nucleus [131]. FP NPs effectively degrade mutp53, counteract cisplatin-induced mutp53 elevation, and activate wild-type p53 expression, inducing endoplasmic reticulum stress, mitochondrial and DNA damage, and ultimately triggering apoptosis in both mutp53- and wtp53-expressing cells. In vivo and in vitro studies confirm that FP NPs significantly suppress tumors, reduce recurrence, and inhibit metastasis, highlighting their potential as a powerful therapeutic strategy for mutp53-driven cancers. These advancements underscore the value of nanocarriers and combination therapies in overcoming chemotherapy resistance and improving treatment outcomes.

#### 3.2.3. Nanoformulations with Intrinsic Mutant p53 Clearance Ability

Mitochondria-targeting compounds have emerged as promising therapeutic agents in cancer treatment, particularly for degrading mutp53 and inhibiting KRAS signaling pathways (Table 4). HA-TPP, a self-assembled nanocarrier composed of hyaluronic acid (HA) and lipophilic alkyltriphenylphosphonium (TPP), targets mitochondria and promotes mutp53 degradation, showing synergistic effects when combined with KRAS inhibitors like AMG510 in KRAS/TP53-mutated cancers [132]. Similarly, AIE-Mit-TPP, a mitochondria-targeting aggregation-induced emission material, enhances mutp53 clearance and induces mitochondrial damage [133]. P^•+^-DPA-Zn, a zinc-based compound generating stable cationic radicals, accumulates in mitochondria, induces oxidative stress, and facilitates mutp53 degradation via the UPS, selectively targeting mutp53 cancer cells [122]. Additionally, P6@siKRAS, a dual-functional nanoparticle combining a pyrrole radical cation (P6^•+^) with KRAS siRNA, simultaneously degrades mutp53 and suppresses oncogenic KRAS, demonstrating potent therapeutic efficacy in pancreatic cancer [134]. These mitochondria-targeting strategies highlight the potential for synergistic cancer therapy, though further research is needed to fully elucidate the relationship between mitochondrial damage and mutp53 degradation. Zhang et al. [135] reported that CeO_2_ nanoparticles (CeO_2_ NPs) can induce the broad-spectrum K48 UPS-mediated degradation of the mutp53 protein. The dissociation of mutp53 from heat shock protein Hsp90/70 has been shown to correlate with the degradation of mutp53. The clearance of mutp53 by CeO_2_ NPs abrogates the GOF exhibited by mutp53, which in turn leads to decreased cellular proliferation and migration, thereby significantly enhancing the therapeutic effectiveness for the BxPC-3 mutp53 tumor. Notably, PEGylated CeO_2_ nanoparticles exhibited a selective clearance of mutp53 and demonstrated cytotoxicity toward p53-mutated cancer cells, highlighting a promising approach to address the challenges associated with mutp53 degradation.

#### 3.2.4. Biomimetic Nanoreceptor for a Specific Degradation of mutp53

Autophagy receptor proteins can bind to cargo ubiquitination signals via ubiquitin-binding domains, while the LC3-binding region associates with ATG8/LC3/GABARAP (LC3) proteins. This interaction facilitates the transport of proteins or damaged organelles to autophagosomes, which subsequently fuse with lysosomes for degradation [63,136]. Huang et al. [137] introduced a multifunctional biomimetic nanoreceptor (NR) designed in accordance with the biological principle governing the selective degradation of cargo by autophagy receptor proteins. These engineering NRs were synthesized through the self-assembly process involving maleimide polyethylene glycol polylactic acid (Mal-PEG-PLA) and 1,2-diol-3-trimethylammonium propane (DOTAP) to form nanoparticles, which were subsequently covalently modified with mutp53-binding peptides (MBPs) to mimic biomimetic selective autophagy receptors (Table 4). The MBPs affixed to the surface of the NRs are capable of capturing and recognizing the mutated p53 protein for degradation, while cationic lipid DOTAP enhances autophagy levels and facilitates the incorporation of NR-bound mutp53 into autophagosomes. The biomimetic NRs have demonstrated the capacity to induce the autophagic degradation of oncogenic mutp53, exhibiting significant therapeutic efficacy in both cellular models and patient-derived tumor xenograft models. Furthermore, the researchers noted that the combination of the cisplatin prodrug with the NR resulted in an amplified cytotoxic effect on cancer cells. It is important to highlight that while the binding of the NR to mutp53 does not require the ubiquitination of the mutant proteins, the subsequent protein degradation is dependent on ubiquitination. NRs exhibit selective binding to different variants of mutp53 proteins, while they do not interact with the wtp53 protein, which is known for its anticancer properties, nor with the p53 family homolog, p63. This pronounced selectivity mitigates potential safety risks associated with nonspecific degradation and underscores the extensive clinical applicability of NRs.
biomolecules-15-00763-t004_Table 4Table 4Nanoformulations for triggering mutp53 clearance.Biomaterial TypesAction on mutp53In Vitro/In VivoMechanismReferencesZnFe-4 nanoparticlesSelective degradation of mutp53In vitro/in vivoInduce degradation of mutp53 via UPS; reduce cell proliferation and cell migration[123]ZIF-8; ZIF-8 modified with Z1-RGD peptidesSelective degradation of mutp53In vitro/in vivoInduce degradation of mutp53 by UPS and glutathionylation-dependent proteasome; reduce the GSH: GSSG ratio[124]Mn-ZnO_2_ nanoparticlesClearing mutp53 and enhancing wtp53 expressionIn vitro/in vivoInduce degradation of mutp53 by UPS, increase ROS level, and activate the ATM-p53-Bax pathway to elevate the wtp53 level[125]Zn-LDH@GOXDegradation of mutp53In vitro/in vivoElevate intracellular Zn^2+^ concentration, promote the transformation of part of mutp53 conformation into wtp53 conformation, reactivate the function of wtp53, and promote the degradation of mutp53 via the autophagy pathway[126]PEGylated CeO_2_ NPsDegradation of mutp53In vitro/in vivoIncrease the production of ROS, promote the degradation of mutp53, and reduce cell proliferation and migration[135]Black phosphorus nanosheetsDegradation of mutp53In vitro/in vivoReduce resistance of tumor cells to chemotherapy drugs and degrade mutp53 protein[138]Crizotinib nanomicellesDegradation of mutp53In vitro/in vivoInduce degradation of mutp53 via UPS, abrogate mutp53-manifested GOF, and reduce cell proliferation, migration, and cell cycle arrest[130]Fluplatin@PEG-PE nanoparticlesDegradation of mutp53In vitro/in vivoDegrade mutp53, trigger endoplasmic reticulum stress (ERS), and mitigate cisplatin resistance caused by mutp53[131]MBP-NPs-DOTAPClearing mutp53In vitro/in vivoElevate the levels of autophagosome formation; increase the degradation of mutp53[137]HA-TPP/ADegradation of mutp53In vitro/in vivoInhibit the signaling pathways of mutant KRAS and mutp53; degrade mutp53 proteins[132]P6@siKRASDegradation of mutp53In vitro/in vivoTrigger UPS-mediated degradation of mutp53, inhibit KRAS signaling pathways, eliminate mutp53’s GOF effects, and suppress tumor growth[134]


### 3.3. Targeting Delivery of Synthetic Lethality Compounds

Given the critical role of synthetic lethality in treating p53-mutated tumors, an increasing number of nanotechnology-based approaches have been developed for the targeted delivery of the synthetic lethal drug adavosertib to overcome cancer resistance. For example, Yu et al. [139] developed estrone-targeted avosertib MOFs that possess sonodynamic therapeutic properties and pH responsiveness. This nanopreparation facilitates rapid endocytosis into cells via estrogen receptors, which are abundantly expressed on gallbladder cancer cell surfaces. Upon entering the acidic TME, the formulation releases adavosertib (ADA), thereby inhibiting DNA repair and inducing synthetic lethality. The application of ultrasound further accelerates ADA release and generates substantial reactive oxygen species, exacerbating DNA damage and enhancing the synthetic lethal effect of the WEE1 inhibitor on p53 mutant cells. Additionally, this nanoformulation significantly mitigates drug toxicity. Importantly, it also demonstrates comparable antitumor efficacy in other solid tumors harboring p53 mutations, offering a novel insight into the utilization of MOF-based drug delivery systems for the treatment of a range of refractory cancers through the conditional synthetic lethality (Figure 7).

### 3.4. Nano-Enabling p53-Based Genetic Therapies

#### 3.4.1. Nano-Enabling p53-Based Gene Therapies

Nano-enabling p53-based gene therapy has garnered significant attention due to its low immunogenicity, which reduces immune-related adverse effects and prolongs the circulation time of the drug in vivo. This therapy specifically enables the selective restoration of p53 function in cancer cells, demonstrating potent anticancer activity in preclinical models such as hepatocellular carcinoma and breast cancer (Table 5). Unlike current cancer gene therapy drugs like Gendicine, which are limited to intratumoral injection, nano-enabling p53-based gene therapy supports intravenous administration, thereby offering superior therapeutic effects against distant metastases [140,141,142]. A significant illustration of this is SGT-53, a cationic liposome that encapsulates wtp53 DNA. This formulation is specifically directed toward tumor cells through the use of an anti-transferrin antibody fragment. SGT-53 has exhibited encouraging outcomes, including the sensitization of glioblastoma cells to temozolomide in vitro and in vivo, an enhancement of survival rates in mouse models, and the demonstration of clinical efficacy in a phase I trial. In this trial, involving eleven advanced solid tumor patients, SGT-53 effectively delivered TP53 to metastatic sites, resulting in stable disease in seven patients at 6 weeks. Ongoing clinical trials are currently assessing its therapeutic potential further.

Similarly, Wu et al. [143] utilized chondroitin sulfate (CS) to modify amine-terminated silica nanoparticles, which possess the capability to target the CD44 receptor located on the surface of tumor cells. These nanoparticles were designed to co-encapsulate near-infrared fluorescent dyes, specifically methylene blue (MB), along with p53 plasmids, leading to the development of MB-NSi-p53-CS ternary complexes aimed at the targeted imaging of tumors and the application of p53 gene therapy in the treatment of lung cancer. The findings indicated that the incorporation of CS-coated polymers maintained the surface charge of these complexes and significantly improved the specific tumor cellular internalization of these complexes. When compared to the positive control, which consisted of PEI 25 K/p53 complexes, these MB-NSi-p53-CS ternary complexes exhibited a markedly superior capacity for delivering the p53 gene into tumor cells, which led to the expression level of p53 mRNA in tumor cells containing mutp53 increases of approximately 1.5-fold. Moreover, this delivery also contributed to higher antitumor efficacy. Misra et al. [141] developed Chol-5LD nanoliposomes, composed of cationic gemini cholesterol (Chol-5L) and dioleoylphosphatidylethanolamine, to facilitate the delivery of the p53-EGFP-C3 fusion genes. These fusion genes consist of the pEGFP-C3 and pCep4-p53 plasmids, which encode for the green fluorescent protein (GFP) and p53 proteins, respectively. The findings indicated that Chol-5LD nanoliposomes effectively mediated the expression of both the GFP gene and the functional p53 gene within tumor cells, demonstrating high transfection efficiency and the ability to induce apoptosis. Consequently, this resulted in the successful induction of p53-mediated apoptosis in the tumor cells.

The concurrent administration of p53 plasmids alongside chemotherapy agents via nanoparticles represents a promising therapeutic approach for challenging malignancies (Table 5). Davoodi et al. [144] synthesized an amphiphilic graft copolymer, PEI-SS-PCL-SS-PEI, utilizing poly(ε-caprolactone) diol (PCL-diol) and low-molecular-weight polyethyleneimine (LMw-PEI). This copolymer was employed to co-deliver Dox and p53 DNA plasmid into target cells. The reactivation of the p53 gene pathway markedly sensitized cancer cells to chemotherapy, promoted a higher rate of tumor cell apoptosis, and more effectively inhibited the migration of cancer cells. Shen et al. [145] developed hollow mesoporous silica nanospheres (HMSNs) by coating their surface and subsequently adsorbing the p53 plasmid onto the nanospheres using polyethyleneimine (PEI). Concurrently, they encapsulated the proteasome inhibitor bortezomib (BTZ) within the nanospheres, resulting in the formation of HMSNs-PEI-BTZ-p53 nanoparticles aimed at treating p53-mutated NSCLC. The findings indicated that these nanospheres effectively facilitated the simultaneous introduction of both genetic material and chemotherapy agents into NSCLC cells, thereby enhancing the bioavailability of the drug and significantly increasing its cytotoxic effects against NSCLC. Furthermore, the application of HMSNs-PEI-BTZ-p53 led to the restoration of p53 expression and the reactivation of the p53 signaling pathway, which collectively contributed to a pronounced synergistic effect in tumor suppression and improved the therapeutic efficacy of BTZ. Chen et al. [146] utilized the polymer CD-PGEA to modify gold nanorods and a mesoporous silica shell composed of bell-shaped rough nanocapsules (Au@HSN-PGEA, AHPs) for the co-delivery of the drug sorafenib (SF) and gene p53 (SAHP/p53), aiming to achieve complementary gene/chemotherapy/photothermal therapy. Experimental results indicated that the nanomaterials triggered the release of sorafenib (SF) under near-infrared light irradiation, enhancing the chemotherapy effect with photothermal therapy.
biomolecules-15-00763-t005_Table 5Table 5Nanostrategies for p53-based gene therapies.Biomaterial (Carrier)CellAnimal ModelsMechanismReferencesMB-NSi–p53-CS ternary complexesA549 cellsMale BALB/c nude mice of 5 weeksElevate expression level of p53 mRNA in tumor cells harboring mutp53; enhance antitumor efficacy[143]f-SWCNTs-p53 complexesMCF-7 cells/Increase p53 mRNA levels in the cells and induce apoptosis[147]DOX/p53 mRNA complexesHepG2 cells, Hela cells, and C6 cells/Activate the p53 pathway; increase the sensitivity of tumor cells to chemotherapeutic drugs[144]HMSNs-PEI-BTZ-p53 nanoparticlesCRL-5872 cells/HMSNs-PEI-BTZ-p53 nanoparticles[145]GOAS-pEFGP-p53 complexesMCF-7 cells; BT-20 cells/Activate the p53 pathway by transferring therapeutic agents into tumor cells; induce cell apoptosis[148]p53-EGFP-C3 fusion constructHeLa cells, H1299 cells, and HEK 293T cellsHealthy nude mice of 5–6 weeksEnhance the expression of the p53 protein; induce cell death[141]TK-PEI/HAP/DNA NCsB16F10 cellsMale C57BL/6 miceRestore p53 expression, increase the generation of ROS, and strengthen the delivery of p53 genes[149]SAHP/p53HEK 293 cells; HepG2 cellsBALB/C nude miceTrigger photothermal treatment, achieve the synergistic effects of photothermal therapy and chemotherapy, and enhance the joint delivery of sorafenib and p53[146]PEN-p53HeLa cells; PC-3 cells/Up-regulate the expression of the p53 gene, inhibit cell proliferation, and activate cell apoptosis and cell cycle arrest[150]Chol-g-PMSC-PPDL/p53 nanoparticlesPC-3 cells/Increase the cellular expression level of p53, inhibit cell proliferation, and activate mitochondria-dependent apoptotic pathways and cell cycle arrest[151]p53/C-rNC/L-FAMCF-7 cellsFamale BALB/c-nu mice, male BALB/c mice, female Sprague Dawley ratsDelivery of CytoC to the cytoplasm and the p53[152]AP-PAMAM/p53HeLa cells/Enhance the expression of the p53 gene, inhibit cell proliferation, and activate apoptosis and cell cycle arrest[153]


#### 3.4.2. Restoring wtp53 Expression Through mRNA Nanodelivery

The optimal delivery vector for RNA should facilitate its entry into the cytoplasm while providing protection during both the entry and transport processes. Additionally, it must address the intrinsic hydrophobic properties and the negative charge associated with mRNA. Therefore, non-viral nanocarriers composed of lipids, polymers, or lipid–polymer hybrids have been widely used (Table 6). Most of these nanocarriers possess the ability to target specific cell types or be eliminated by cellular mechanisms over time. These properties contribute to a reduction in systemic toxicity and the accumulation of carrier molecules, thereby minimizing the overall toxicity of such therapies. Recent reviews by Huang et al. have discussed the latest advancements in mRNA nanomedicine [154].

Recent advancements show that redox-responsive nanoparticles carrying p53 mRNA effectively reduced viability in p53-null lung cancer cells and significantly decreased tumor size in HCC and NSCLC mouse models. Notably, the antitumor effect was enhanced when combined with mTOR inhibitors. In HCC models, the combination of p53 mRNA nanoparticles with immunotherapy yielded superior anticancer effects compared to monotherapies, suggesting synergistic potential with the immune checkpoint blockade. These findings indicate a promising direction for combination therapies.

Wtp53 mRNA delivered via nanocarriers can achieve targeted delivery to tumor cells, controlled drug release, and reduced systemic toxicity and have independent or combined therapeutic efficacy [155,156]. Recent advancements show that redox-responsive nanoparticles carrying p53 mRNA effectively reduced viability in p53-null lung cancer cells and significantly decreased tumor size in HCC and NSCLC mouse models [157]. Similarly, Dong et al. [158] developed paclitaxel amino lipid (PAL) nanoparticles designed to transport the chemotherapeutic agent paclitaxel while simultaneously encapsulating mRNA encoding p53. Compared to clinically used formulations such as Abraxane^®^ and Lipusu^®^, the PAL nanoparticles exhibited enhanced characteristics, including increased drug loading capacity and improved encapsulation efficiency for both paclitaxel and mRNA. Notably, the PAL nanoparticles containing p53 mRNA demonstrated a synergistic cytotoxic effect in triple-negative breast cancer cells, combining the therapeutic effects of both paclitaxel and p53 mRNA. Furthermore, these chemotherapy drugs-derived nanoparticles encapsulating p53 mRNA displayed significant antitumor efficacy in the mouse models of orthotopic triple-negative breast cancer xenografts. Zhou et al. [159] developed ROS-responsive oligomer-based polymeric NPs (dihydrolipoic acid, o-DHLA) to deliver p53 mRNA and indocyanine green (ICG), demonstrating synergistic therapeutic effects for lung cancer. Tang et al. [160] utilized G0-C14 cationic lipids, mRNA hyaluronic acid, and DSPE-PEG-mannose to develop inhaled, dual-targeted mRNA nanoparticles, successfully targeted to deliver p53 mRNA to lung cancer cells and inflammatory macrophages within the lung. Additionally, Park et al. constructed a fusion protein of apolipoprotein and a targeted antibody to functionalize mRNA–lipid nanoparticles for targeted mRNA delivery. The synthesized trastuzumab-bound p53 mRNA–lipid nanoparticles alleviated the liver toxicity of the antibody-free mRNA–lipid system, enhanced the targeting of Her-2-positive tumor cells, and exhibited superior therapeutic potential [161]. Xiao et al. [162] developed lipid nanoparticles composed of G0-C14, PLGA, and lipid PEG, which were modified with the targeting peptide CTCE-9908. This peptide is specific to CXCR4, a receptor that is highly expressed in liver cancer cells, thereby facilitating the targeted delivery of these nanoparticles to liver cancer cells. Using this novel lipid nanoparticle as a delivery carrier, p53 mRNA can be specifically delivered to liver cancer cells, and when combined with anti-PD-1 monoclonal antibodies, it can significantly induce the global reprogramming of TME, thereby enhancing antitumor efficacy.

However, when considering the DN and GOF activity of mutp53, the mere supplementation of cells with wtp53 may prove inadequate for the effective induction of wtp53-mediated tumor suppression. To optimize the functionality of introduced or restored wtp53, it is essential to deplete mutp53. This can be accomplished through the utilization of established pharmacological agents, such as statins, or by the application of small RNAs specifically targeting mutp53, such as siRNA. The introduction of wtp53 into cells deficient in mutp53 can augment its activity to the necessary threshold required to suppress tumor cell proliferation. This enhancement may subsequently improve the likelihood of delaying malignant progression and/or preventing cancer cells from developing resistance to therapeutic agents.
biomolecules-15-00763-t006_Table 6Table 6Nanostrategies for restoring wtp53 expression through mRNA nanodelivery for cancer therapy.Biomaterial (Carrier)CellAnimal ModelsMechanismReferencesp53 mRNA–lipid nanoparticles (p53 mRNA@LNPs)MDA-MB-231, SK-OV-3, MDA-MB-453, SK-BR-3 cells, and BT-474 cells7-week-old female BALB/c nu/nu miceImprove the expression levels of p53; induce cell death in a dose-dependent manner[161]DNA nanoparticlesRAW264.7 cells; DC2.4 cells8-week-old male BALB/c JGpt miceImprove mRNA vaccine delivery and efficacy, activate immune responses, and induce the production of antigen-specific antibody[155]Lipofectamine Messenger MAX Transfection ReagentHuman ovarian cancer cell lines (SKOV3, OVCAR-3, OVCAR-4, OVCAR-5, OVCAR-8)6–8-week-old female nude miceLipofectamine Messenger MAX Transfection Reagent[156]A ROS-responsive polymeric nanoparticleH1299 cellsAthymic nude micePromote mRNA translation efficiency and p53 expression; induce generation of ROS[159]PLGA/lipid–PEG/lipid nanoparticlesHCC cells, RIL-175 cells, and HCA-1 cells5–6-week-old or 6–8-week-old immunocompetent male and female C57BL/6 miceIncrease p53 expression, restore p53 functional activity, reduce cancer cell viability, and inhibit tumor growth improved tumor sensitivity to immunotherapies[162]HA/DSPE-PEG/mannosenanoparticlesH1299 cells; HCT116 cells4–6-week-old female BALB/c mice; 6-week-old C57BL miceDeliver the targeted p53 proteins into lung tissues, accumulate p53 mRNA in lung tumor cells and inflammatory macrophages, and enhance the expression of p53 proteins[160]Redox-responsive polymer PDSA/DSPE-PEG/DMPE-PEG nanoparticlesHep3B cells; H1299 cells4–6-week-old female athymic nude mice, 6-week-old wild-type BALB/c mice, and 4-week-old female C57BL/6 miceRestore p53 function, impede the proliferation of p53-deficient liver and lung cancer cells, induce cell cycle arrest and apoptosis, reverse the resistance of cancer cells to the mTOR inhibitor[157]Paclitaxel amino lipid (PAL) nanoparticlesMDAMB-231 cellsathymic nude female miceEnhance encapsulation efficiency for both paclitaxel and mRNA[158]PBA-BADP/mRNA nanoparticlesHeLa cells, SiHa cells, DU145 cells, CCC-HPF-1 cells, HK-2 cells, and HEK293 cells/Selectively prohibit cancer cell growth[163]PRIZE, a p53-repair nanosystem4T1 cells, MC38 cells, Luc-4T1 cells, 4T1-OVA cells, and MC38-OVA cellsBALB/C mice; C57BL/6 miceRestore intracellular p53 levels; trigger immunogenic cell death[164]


#### 3.4.3. Ablation of p53 Expression by Genetic Approach

The clinical utilization of siRNA is impeded by several challenges, including low transfection efficiency, inadequate tissue penetration, and unintended immune activation. A critical concern in siRNA-based tumor therapy is the precise and efficient delivery of siRNA to both primary and metastatic tumors. Kundu et al. formulated a lipid–polymer hybrid nanoformulation capable of delivering siRNA to effectively silence mutp53 in mouse osteosarcoma cell lines [165]. Eduardo et al. [166] utilized low-molecular-weight branched polyethylenimine (bPEI) to modify the synthesized gold nanoparticles (AuNPs) to enhance the delivery efficacy of gapmers aimed at the mutp53 protein. Furthermore, Yoon et al. [167] investigated the mutated forms of the EGFR and TP53 oncogenes, specifically the single-nucleotide missense mutations EGFR-T790M and TP53-R273H, which are implicated in resistance to gefitinib. The simultaneous delivery of an adenine base editor (ABE) along with a single-guide RNA targeting the EGFR and TP53 SNPs through adenoviral (Ad) led to the accurate and efficient correction of oncogenic mutations, both in vitro and in vivo. Significantly, in comparison to a control group that received treatment solely with gefitinib, an EGFR inhibitor, the concurrent administration of Ad/ABE targeting SNPs in TP53 and EGFR, in conjunction with gefitinib, resulted in an enhanced drug sensitivity and a more effective suppression of abnormal tumor proliferation (Table 7).

### 3.5. Nanodelivery of wtp53 Proteins

One innovative approach involves the utilization of stimuli-responsive nanocarriers to selectively target cancer cells for the delivery of p53 proteins (Table 8). Tang et al. [168] engineered luteinizing hormone-releasing hormone (LHRH) peptide-conjugated nanocapsules that effectively facilitated the targeted delivery of recombinant human p53 proteins to LHRH receptor-expressing tumor cells, resulting in the reactivation of apoptosis in these cells. Similarly, Guan et al. [169] utilized extracellular vesicles (EVs) obtained from breast cancer cells to facilitate the targeted delivery of the exogenous p53 protein, which was modified with triphenylphosphine (TPP). The EV carriers exhibited a specific affinity for the originating tumor cells and exhibited the ability to penetrate the mitochondrial membranes of these cells. Exogenous p53 could promote apoptosis in breast cancer cells via the Bcl-2/Bax apoptosis signaling pathway. In vivo experiments revealed that the exogenous p53 protein-loaded EVs modified with TPP predominantly accumulated in tumor cells, exhibiting no toxic side effects on normal tissues and organs. Furthermore, A notable decrease in both the volume and mass of tumors was observed following the administration of the p53 nanocarriers. These findings offer significant insights that could guide the development of innovative and effective p53 nanocarrier systems for the treatment of various tumors. However, mutations in the p53 gene are the most prevalent single-gene alterations observed in human cancers, which impede the anticancer functions of wtp53. Therefore, further investigation is warranted to determine whether enhancing the delivery of the p53 protein is an ideal approach for the treatment of cancers characterized by p53 mutations.

## 4. Concluding Remarks and Future Directions

p53 mutations are highly heterogeneous and diverse [173], and different mutants may require distinct therapeutic strategies. Second, the p53 protein lacks well-defined binding pockets (except for the Y220C mutation), and the mechanisms underlying its reactivation remain unclear, complicating drug design. Furthermore, resistance to p53 mutations, the off-target effects of drugs, and the potential toxic side effects further increase the difficulty of treatment. The structural details of full-length p53 in association with DNA targets or interacting partner proteins are not fully elucidated, limiting structure-based drug design. Nonetheless, targeted therapeutic strategies for p53 are continuously evolving and improving. First, restoring the function of mutp53 is a key direction. Although challenging, small-molecule compounds that facilitate the restoration of the wild-type conformation and function of mutp53 offer the advantage of targeting mutp53, which is typically overexpressed and present only in tumor cells. Second, with the continuous exploration of the mechanisms of maintaining the stability of mutp53 proteins, regulating the degradation of oncogenic mutp53 remains a promising approach for treating p53-mutated tumors. Due to the approval of a drug for delivering the wtp53 gene, p53-based genetic therapies are a research direction. Additionally, novel therapeutic technologies like PROTACs, CRISPR-Cas9 gene editing, antibody–drug conjugates (ADCs), and engineered cell therapies (e.g., CAR-T cells) provide new tools and methods for p53-targeted therapy [13].

However, based on experimental and clinical findings, several currently promising therapeutic aroaches for p53-mutated cancers still face significant challenges. The clinical data on APR-246, ATO, and PC14586 highlight both the promise and challenges of targeting TP53-mutated cancers. APR-246 demonstrates efficacy in hematologic malignancies when combined with azacitidine, particularly in improving complete remission rates, though its survival benefit remains ambiguous due to increased mortality in some trials. In solid tumors like ovarian cancer, while deep responses are observed, toxicity and inconsistent disease control pose significant limitations. ATO shows preclinical potential in restoring p53 function, but its clinical applicability is restricted by narrow therapeutic windows and exceedingly rare targetable mutations. PC14586, a Y220C-specific agent, exhibits manageable safety and preliminary activity, yet its broader utility awaits further validation. Additionally, it should be noted that the effectiveness of p53 gene therapy likely depends on the TP53 status within cancer cells. While p53-null tumors and those with truncating or frameshift TP53 are ideal targets, tumors with TP53 missense mutations present a challenge. Although they may benefit from wtp53 restoration, the dominant-negative effect of accumulated mutp53 could counteract the therapy. A meta-analysis examining patients with recurrent HNSCC who received adenovirus–p53 gene therapy revealed that those who responded to the treatment exhibited low levels of p53 staining and lacked TP53 missense mutations [174], suggesting that high mutp53 levels may limit efficacy. Strong p53 immunohistochemistry staining could thus serve as an exclusion criterion. However, more clinical data are needed to confirm the long-term potential of wtp53 gene therapy. Since targeting only p53 may not be sufficient to cure cancer, combination therapy has become a critical strategy. These therapies underscore the need for biomarker-driven patient selection, optimized combination strategies, and novel delivery systems to balance efficacy and toxicity. Future success hinges on resolving these challenges through precision approaches and robust clinical trial designs.

Nanotechnology offers new tools and methods for p53-targeted therapy, demonstrating significant advantages. First, nanoparticles can achieve precise targeted delivery through surface modifications, delivering drugs directly to tumor cells expressing mutp53 while minimizing damage to normal tissues. Second, nanocarriers can protect drugs (such as mRNA or siRNA) from degradation, prolonging their circulation time in the body and enhancing their bioavailability and efficacy. Additionally, nanotechnology’s multifunctionality allows for the simultaneous delivery of drugs, genes, and imaging agents, enabling the combination of mutp53 targeting with existing cancer therapies and facilitating theranostic applications. Interestingly, recent studies have revealed that specific nanomaterials, such as PEGylated CeO_2_ nanoparticles [135] and mitochondria-targeting nanoformulations [133,134], exhibit a unique capability to selectively degrade mutp53 without affecting wtp53. This inherent selectivity opens up new avenues for targeted therapeutic interventions in mutp53-driven cancers. Particularly, mitochondria-targeting strategies underscore the potential for synergistic cancer therapy by leveraging the interplay between mitochondrial dysfunction and mutp53 degradation. However, the precise mechanisms linking mitochondrial damage to mutp53 clearance remain to be fully elucidated, warranting further investigation to optimize these promising therapeutic approaches. Furthermore, emerging technologies such as PROTAC and the innovative concept of biomimetic nanoreceptors have been proposed as novel platforms for targeting the degradation of mutp53. These technologies can precisely identify abnormally expressed mutp53 in tumor cells and achieve efficient degradation by mimicking the properties of natural receptors, offering a promising new direction for mutp53-targeted therapy.

However, these studies are currently primarily at the preclinical stage, and there are still many challenges to overcome before practical application. Although the introduction of nanotechnology has improved drug bioavailability through targeted delivery and controlled-release properties, its application remains constrained by two major contradictions: the idealized design of precise delivery versus the reality of complex biological barriers in vivo. For instance, the tumor accumulation efficiency of liposomes or polymeric nanoparticles is affected by the heterogeneity of vascular leakage, while the inherent immunogenicity of nanomaterials may trigger nonspecific inflammatory responses, thereby diminishing therapeutic efficacy. More critically, the lack of standardized protocols for the large-scale production and quality control of nanocarriers hinders their rapid translation from laboratory to clinical settings. Additionally, the heterogeneous expression and functional variability of mutp53 further complicate the development of treatments. Moving forward, it will be necessary to conduct large-scale animal experiments and translational research to systematically assess the pharmacokinetics, toxicity, and efficacy of nanomaterials, as well as to explore their combined use with traditional therapies. Future investigations should focus on several key directions. First, precision medicine will play a pivotal role. Functional assays and ex vivo screening can identify patients most sensitive to p53-targeted therapies, enabling personalized treatment. Second, the development of novel nanomedicines is crucial. Creating non-immunogenic, liver-evading, and tissue-targeting non-viral delivery particles will further improve therapeutic outcomes. Additionally, exploring combination strategies is essential. Combining p53-targeted therapy with other treatments, such as immune checkpoint inhibitors, may yield synergistic effects and enhance therapeutic efficacy. Finally, in-depth basic research will provide theoretical support for p53-targeted therapy. Investigating the interactions between p53 and other tumor-related genes and signaling pathways will aid in developing more effective p53-targeted therapies.

Although p53-targeted therapy faces numerous challenges, significant progress is being made in p53 drug development with advancing technologies. The utilization of nanotechnology, gene editing, and innovative drug delivery systems has opened new possibilities for p53-targeted therapy. In the future, with the advancement of precision medicine and personalized treatment, p53-targeted drugs are expected to bring breakthroughs in cancer therapy, offering more effective treatment options for patients.

## Figures and Tables

**Figure 1 biomolecules-15-00763-f001:**
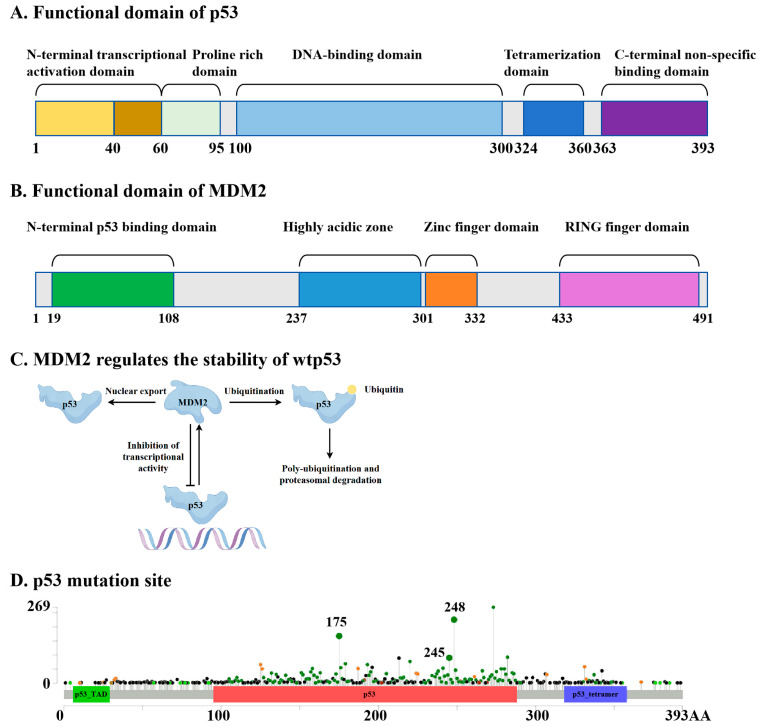
Schematic diagram of the functional domains of p53 (**A**) and MDM2 (**B**). (**C**) MDM2 maintains the functional stability of wtp53 by promoting its degradation and inhibiting its transcriptional activity. (**D**) Specific mutation sites of p53 (the data come from cBioPortal).

**Figure 2 biomolecules-15-00763-f002:**
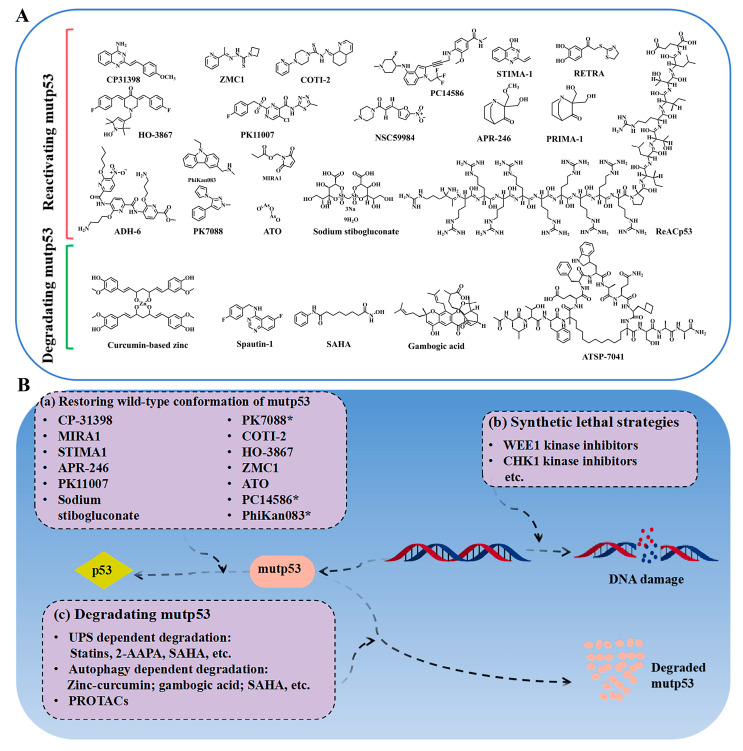
(**A**) The structures of some small-molecule compounds used for p53-mutated tumors; the chemical structures of the molecules were drawn by ChemDraw 22.0.0 64-bit. (**B**) The differential strategies to target p53-mutated tumors, including restoring the conformation of mutp53 by small-molecule compounds, inducing synthetic lethality, and clearing mutp53.

**Figure 3 biomolecules-15-00763-f003:**
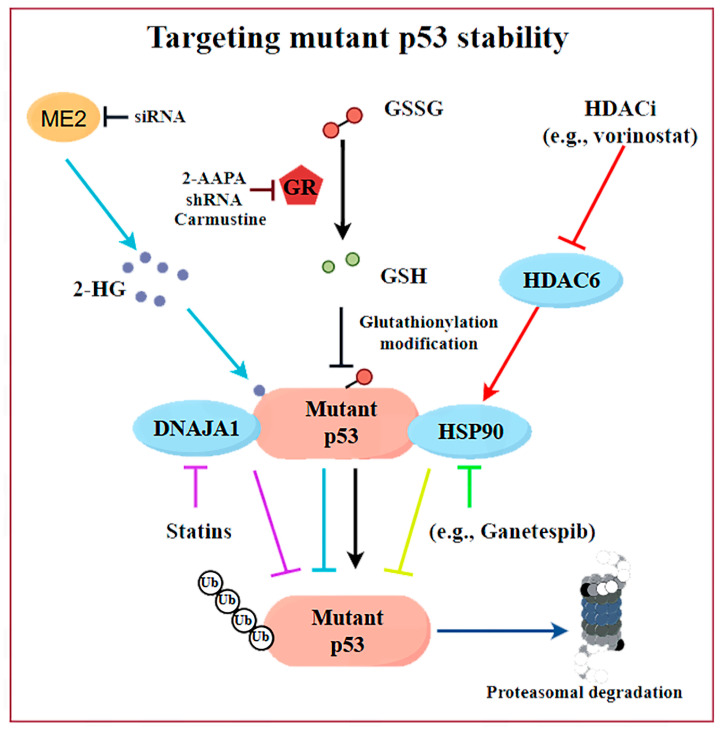
The discovered mechanisms underlying mutp53 protein stability and related intervention strategies. Disrupting the binding of mutp53 to molecular chaperones (DNAJA1 and HSP90), inhibiting 2-HG synthesis, targeting GSH reductase, or depleting GSH would promote the degradation of mutp53. Created by Figdraw.

**Figure 4 biomolecules-15-00763-f004:**
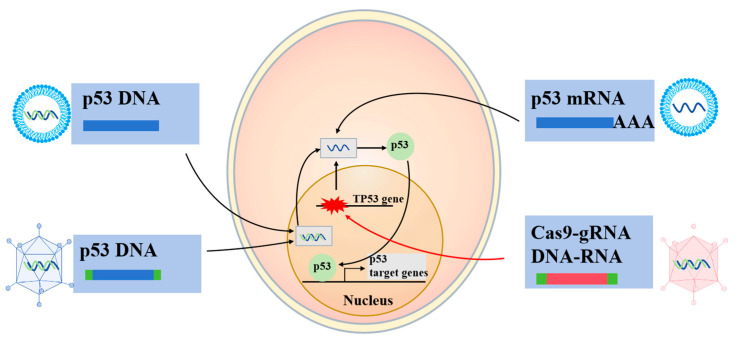
Genetic therapy based on p53. The DNA or RNA encoding wtp53 could be delivered to cancer cells through various pathways, including recombinant viruses and nanoparticles, and promote the expression of wtp53 and the transcription of downstream target genes, thereby producing anticancer effects. In cancer cells with p53 mutations, the delivery of CRISPR-Cas9 along with appropriate guide RNA (gRNA) may achieve base editing and restore the wtp53 sequence. This graph was drawn by Microsoft Office PowerPoint (Office16) with reference to the previous literature [6].

**Figure 5 biomolecules-15-00763-f005:**
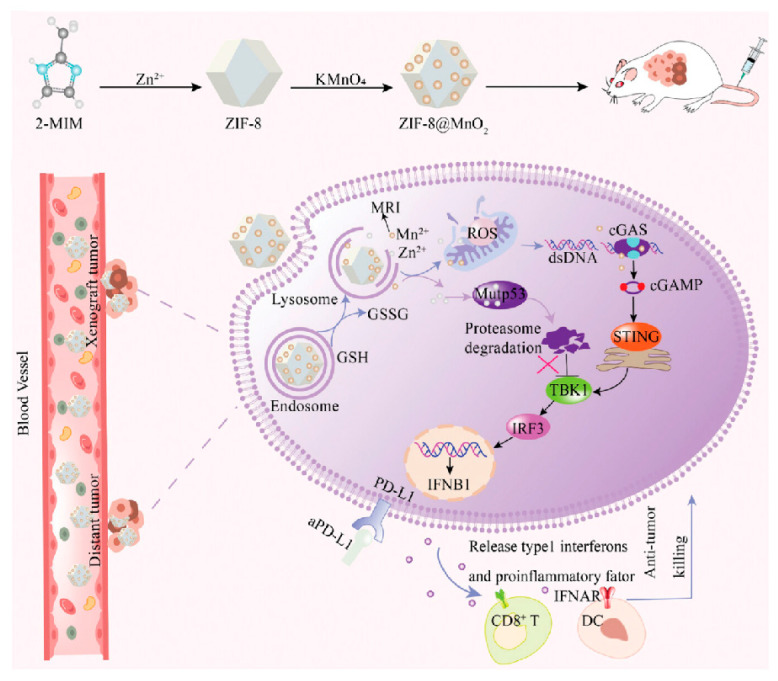
Schematic illustration of the synthesis process of ZIF-8@MnO_2_ nanoparticles and the mutp53 degradation mechanism. Reproduced with permission from [128]. Copyright 2024, Xangpeng Zheng et al. Advanced Science published by Wiley-VCH GmbH.

**Figure 6 biomolecules-15-00763-f006:**
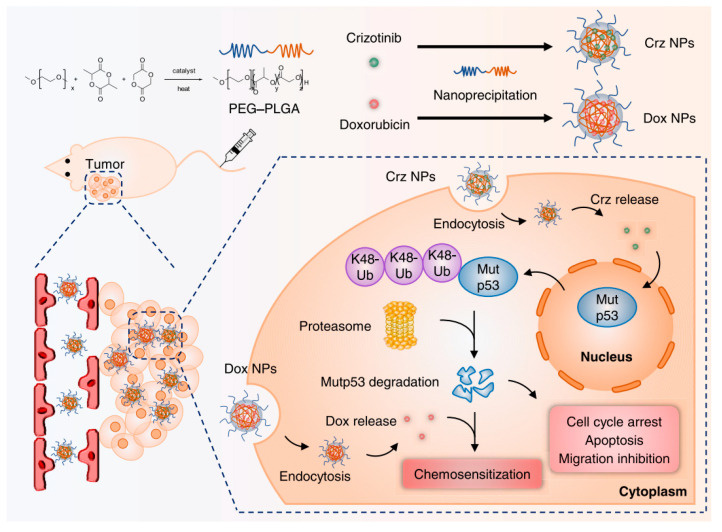
Schematic representation illustrating the synthesis of crizotinib/Dox-loading nanomicelles and the synergistic therapy of Dox and crizotinib for mutp53-driven cancer. Reproduced with permission from [130]. Copyright 2023, American Chemical Society.

**Figure 7 biomolecules-15-00763-f007:**
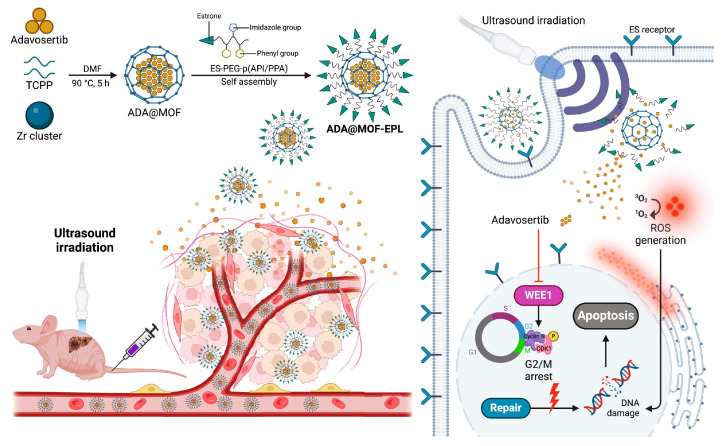
Schematic illustration of the synthesis and the treatment mechanism via synthetic lethality of ADA@MOF-EPL. Reproduced with permission from [139]. Copyright 2024, Science China Pres. Published by Elsevier B.V. and Science China Press.

**Table 3 biomolecules-15-00763-t003:** Nanoformulations for delivering mutp53-reactivating agents.

Biomaterial Types	p53 Reactivating Agents	In Vitro/In Vivo	Mechanism	References
Cer-RUB nanomicelles	C16-ceramides	In vitro/in vivo	Restore the expression of wtp53 in cancer cells or transgenic mice harboring mutp53	[113]
MtrapNPs	ATO	In vitro/in vivo	Rescue p53 mutations and inhibit MDM2	[117]
CD19-targeted polymersome	ATSP-7041	In vitro/in vivo	Reactivate p53 and inhibit the function of the BCL-2 protein family	[118]
ANVs	ZMC	/	Restore the structure and function of wtp53 and restore DNA damage repair	[119]
Thermosensitive gel–nano system	PRIMA-1	In vitro/in vivo	Restore p53 activity and boost T-cell immunity	[120]

**Table 7 biomolecules-15-00763-t007:** Nanostrategies for ablating p53 expression by genetic approach for cancer therapy.

Biomaterial Types	Cell	Animal Models	Mechanism	References
PLGA hybrid lipid	Mouse osteosarcoma cell line carries p53R172H mutant alone	/	Knocked down mutp53 efficiency	[165]
Adenovirus	H1975(EGFR-T790M; TP53-R273H)	6-week-old male athymic nude mice	Increased drug sensitivity; suppressed tumor growth	[167]
Gold nanoparticles (AuNPs) modified with bPEI	PANC-1 (mutp53-R273H); MDA-MB-231 (mutp53-R280K)	/	Enhanced the delivery of gapmers targeting mutp53 protein	[166]

**Table 8 biomolecules-15-00763-t008:** Nanostrategies for delivering wtp53 proteins for cancer therapy.

Biomaterial Types	Cell	Animal Models	Mechanism	References
TPP/P53@EVs	SK-BR-3 cells, MCF-7 cells, and 4T1 cells	6-week-old BALB/c female mice	Induce tumor cell death with no obvious toxicity or side effects in vivo	[169]
Clickable p53 nanocapsules	MDA-MB-231 cells, SK-OV-3 cells, and HFF cells	/	Deliver recombinant human protein p53 to the targeted tumor cells, leading to the reactivation of apoptosis in these cells	[168]
Pos3Aa-p53 protein crystals	4T1 cells	BALB/c mice	Restored p53 activity by delivering the p53 protein to tumor cells; induced anti-PD-1 immunotherapy	[170]
SNCPs@st-p53 peptide	HepG2 cells	5-week-old Balb/c male nude mice	Delivered p53 peptide to tumor tissues and inhibited tumor growth and induced p53-mediated cell apoptosis	[171]
p53-Lzk/LzE-CPP	U251MG cells, T98G cells, LNZ308 cells, and U87ΔEGFR cells	C57BL6 pregnant female mice	Delivered the p53 protein into tumor cells by CPP; inhibited cell-specific proliferation	[172]

## Data Availability

Not applicable.

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
