# Peer review of "Discovery of Drugs Targeting Mutant p53 and Progress in Nano-Enabled Therapeutic Strategy for p53-Mutated Cancers"

_biomolecules, 2025, doi:10.3390/biom15060763_

Round 1
Reviewer 1 Report
Comments and Suggestions for Authors
This is a very comprehensive but well written review by Zhang et al. I have only some minor suggestions. Then it can be published.
- Please introduce all abbreviations, upon their first use in the text and then use them constantly (e.g. GSH, EPR effect, TME). Just introduce abbreviations, if you really use them later on.
- Line 74. Change „Recent“ to „recent“
- Line: 81-84: I do not understand this sentence
- Line 98-101: I would suggest to add a figure with the chemical structure of all these drugs
- Chapter 2.1. I would recommend to add more references of the original publications and also the study numbers of the mentioned clinical trials (clinicaltrial.gov). Just referring to old reviews somehow questions the need of a new one… You stress that Hassin et al have summarized the clinical status of APR-246, but this article is from 2023. Several studies are now completed. You should give an update. If there are no official reports, maybe you could contact the company?
- Line 141 &211: Change “Phase” to “phase”
- - Line 142/143: There is more than on study! Please update!
- The chapter in Coti-2 is written in a confusing manner. In fact, Coti-2 and the ZMC drugs were discovered in independent studies using a completely different approach. This should be worked out more carefully.
- Line 246&247: Again I would suggest to show the chemical structures in a figure
- Line 265: Why do you mention the developing company here but not for other drugs?
- Chapter 2.4.1: Add the study numbers of the mentioned clinical trials (clinicaltrial.gov).
- Chapter 2.4.2: Are these approaches clinically tested?
- Line 404: Are the ZMCs the ones of the Coti-2 chapter? Please specify and better connect these two chapters.
- Line 417-425: It’s a bit late for these statements. Zn is also central in the P53 restoration e.g. of the coti-2 derivatives. Please connect these two chapters better.
- Line 427: Who is ZIF-8?
- Line 456-458. Please add a ref to this statement
- Intro chapter 3.3.: This is in part repetitive to chapter 2.3., Please better connect these two chapters and refer back to already explained things
- Line 560-563: The introduction does not fit into the flow of the review. The chapter before is about nanoformulations not viral vectors. This is confusing. Please improve.
- Line 568: Why do you mention the developing company here but not for other drugs?
- Line 712: Explain what LHRH is, I guess it’s a targeting peptide? This is probably not familiar to all readers
Reviewer 2 Report
Comments and Suggestions for Authors
The review describes current approaches to the treatment of p53 mutated cancers with an emphasis to the nanocapsulated investigational drugs, siRNA, etc. The latter part of the review is very detailed and well illustrated. Some of the nanocapsulated treatments (like Zn) are general, not specific for p53, so it is not clear why they are listed.
However, for a reader not directly working in the field of p53, it would be desirable to see the introduction showing the wild-type p53 stability regulation pathways through recognition by MDM2, to illustrate this with the picture of p53 structure pinpointing mutated residues impacting p53 recognition by MDM2 and interaction with DNA. Otherwise, in its present format the review looks like being written for specialists knowing locations of all mutations in p53 and their role in p53 interactions.
For a general audience, more elaborated introduction is necessary
Reviewer 3 Report
Comments and Suggestions for Authors
The authors aimed to review evidence on drugs targeting mutant p53 and highlight progress in
nano enabled therapeutic strategy for p53-mutated cancers. Undoubtedly, there is growing research interest in therapies targeting mutant p53. A number of relevant reviews have already been published. This manuscript in its present form fails to summarize evidence in a meaningful way, guiding thus future research efforts .
Specific comments:
- The main scope of the review is not defined. In fact, the manuscript presents two different topics. Section 3 should be the objective of another review.
- Tables should be included, presenting the molecules that have been designed, the relative experimental and clinical data and citations.
- The efficacy of the molecules should be reported.
- Safety data should be reported.
- The obstacles of effective therapeutic strategies should be discussed.
- Guidance to future therapeutic strategies should be provided.
- In the concluding session, the authors should focus on the critical interpretation of current evidence.
Round 2
Reviewer 3 Report
Comments and Suggestions for Authors
The manuscript has been improved. The authors have adequately responded to all Author comments. I recommend publication of the manuscript.